# Shared Autonomous Vehicles Competing with Shared Electric Bicycles: A Stated-Preference Analysis

Sungwon Lee [1], Devon Farmer [1], Jooyoung Kim [2] and Hyun Kim [1,*]

1 Transportation ICT Convergence Research Center, Korea National University of Transportation, Chungju 27469, Korea

2 Department of Transportation Planning & Management, Korea National University of Transportation, Uiwang 16106, Korea

* Correspondence: hyunkim@ut.ac.kr

**Abstract:** Understanding the factors that affect the uptake of emerging transport modes is critical for understanding if and how they will be used once they are implemented. In this study, we undertook a stated-preference analysis to understand the factors that affect the use of shared autonomous vehicles and shared personal mobility (micromobility) as competing modes on a university campus in Korea. We applied a binary logit model, which included time and cost variables as well as the perceptions of convenience (in-car congestion and availability) and safety. For autonomous vehicles, the cost- and time-related demand elasticities were estimated to be −0.45 and −0.25, respectively, while the cost elasticity for shared electric bicycles was −0.42. The elasticities of perceived convenience (availability) and safety for the shared electric bicycle system were estimated to be 0.72 and 0.29, respectively. Finally, the elasticity for perceived convenience (in-car congestion) of the shared autonomous vehicle was 0.42. Our results show that there is an innate preference for shared autonomous vehicles when these are compared to shared personal mobility, and that the effect of subjective variables (convenience and safety) on the use of emerging transport modes is as important as traditional cost and time variables.

**Keywords:** stated preference; shared autonomous vehicles; electric bicycles; shared transport; emerging transport

## 1. Introduction

In this paper, we focus on two promising and emerging means of transportation: shared autonomous vehicles (SAVs) and shared personal mobility (SPM). The term SAV generally refers to relatively small, automated road vehicles (ranging from taxis/cars to small buses or 'shuttles'), which provide on-demand or fixed-route services. SAVs differ from so-called driverless or robo-taxis and transportation network company (TNC) services (e.g., Uber) in that they are a form of public transportation, because they are shared with other users [1]. In this paper, we use the term SPM for small and lightweight vehicles that generally transport users over short distances and are part of a shared service. SPM is sometimes referred to as micromobility, and examples include bike share, electric scooter share, and electric bicycle share etc. [1]. In the case of this study, the SPM that was available included a shared electric bicycle system and a shared electric scooter system. Both SPM and SAV are receiving significant attention around the world [2,3], and thus further understanding of the factors that affect the uptake of these modes, and further understanding of the degree of their impact in advance could have great significance for their successful introduction.

Autonomous vehicles, including those that are shared, enjoy numerous benefits over human-driven vehicles [4–8], including, importantly, reducing road accidents and fatalities [9,10]. Similarly, e-bikes specifically have proven to be an attractive transport

mode in earlier studies [11], and more recent research has shown that e-bike availability can lead to increased cycling and decreased car-dependence [12,13] and health benefits [13], especially in hilly urban environments, since they require less physical effort than conventional bicycles [14,15]. Additionally, shared (but not necessarily electric) bicycles have been extensively studied and have been shown to be generally well used. However, the ability of shared mobility to increase sustainable mode shares and increase access to cycling, has been poorer than it is purported to be [16,17]. Further, SPM, and particularly electric scooters [18–21], have been associated with serious and non-serious accidents.

This study was survey-based, and was undertaken at the campus of the Korea National University of Transportation (KNUT) in Chungju, Korea. Currently, KNUT is home to two popular SPM systems, but due to the risks involved with SPM, university administration would prefer to improve mobility on campus with convenient and safer options, such as SAVs, despite such services and vehicles not yet being commercially available. However, it is unlikely that SPM would be removed; rather, in the future, an SAV service and SPM are likely to be competing modes on this campus. Within the literature, there are few studies directly comparing these two modes as competitors using a stated preference (SP) approach, and therefore we could not fully understand which factors affect the demand for SPM and SAV.

As such, it is important to understand the influences of these two emerging and potentially complementary mobility services on mode choice, and further understand how various aspects of the journeys they provide play a role in selecting each mode, using an SP approach. SP is a methodology for analyzing the effects on transport modes that do not exist or policy variables that are not yet implemented. It assumes a hypothetical market situation and estimates the respondent's utility function through selection data related to various attributes of transportation, and quantitatively estimates the response of transportation users to changes in policy variables from the estimated utility function. In addition to administering the SP questions, the survey asked respondents questions related to their transportation habits, experience with and willingness to use SAVs and SPM, and various attitudinally related questions.

In addition to cost and time-related variables, which are typically used in transportation demand estimation models [22], our estimation model included the perceptions of both comfort/convenience (for SAV and SPM) and safety (for SPM). Scholars and some practitioners have tended to include comfort/convenience and/or safety (especially for bicycles) as variables and/or important considerations more commonly in transport studies [23–27], and this has been the practice for some time [28].

In this paper, we first undertake a review of the relevant literature, with a specific focus on SP studies that have included SAV or AV modes, and then discuss the importance of safe bicycle infrastructure, introduce the reader to the study area and the SAV and shared SPM modes that are available, describe the methodology employed in the analysis, present the results of the study, and finally provide concluding remarks.

## 2. Literature Review

In this literature review we review recent works describing SAVs and SPM, and discuss their potential roles in future transportation networks; we then review recent studies that have undertaken SP surveys relating to autonomous vehicles, especially those focusing on SAVs. The purpose of this review is to understand and summarize how other studies have undertaken SP surveys for autonomous vehicle modes, develop an understanding of what methods they have used, what conclusions and policy implications they have drawn, and finally understand where there are gaps in the literature.

### 2.1. Automated Non-Rail Public Transport

It is predicted that automated vehicle (AV) technology will transform ground-based transportation [2,7,29] and will be applied to all types of existing road-based transport systems, including privately owned vehicles, public transport, and demand-responsive taxi

or ride-hailing services that may or may not be shared (sometimes referred to as robo-taxis). Each of these sub-types of AV will find its place in the transportation market and will appeal differently to different people for different kinds of trips [7].

It is predicted that SAVs of various sizes will be used as complements to or replacements for traditional manually driven and generally fixed-route public transport [30], potentially through the provision of on-demand services [2,29,31]. The automation of non-rail public transport is expected to reduce operating costs and increase network efficiency, therefore allowing for potentially additional service hours and/or routes to be provided [6,29]. In addition to network efficiency, simulations of both privately-owned AVs and of shared AV services have shown that the same or similar benefits can be realized for both buses and personal vehicles, including reducing congestion and emissions [4], minimizing total travel times [6,8], increasing efficiency [5,6], and improving safety [9,10], among other benefits [7].

## 2.2. Stated-Preference Studies Involving AVs

SP surveys and related choice modeling are a commonly employed method for understanding the choice of modes that currently do not exist in transportation networks, such as various AV subtypes. Given that AVs will play an important role in future transportation networks, there have been are numerous mode choice studies for privately owned AVs [32–42], robo-taxis (on-demand driverless taxi or ride-hailing services) [40,43–45], and shared autonomous vehicles/autonomous shuttle buses/automated public transport [30,36,40,46–49].

Ghartzonikas & Gkritzka [50] and Jing et al. [51] have recently reviewed a large number of studies employing stated preference/acceptance for a variety of AV typologies. Ghartzonikas & Gkritzka's [50] review examined studies relating to the determination of the choice of AV modes that were published in reputable journals between 2012 and 2018. Among these studies, eight employed stated-preference surveys and related analyses. They described certain themes within SP studies, including potential barriers for their use (including legal, lability & ethics, security, data privacy, learning curves, environmental concerns, and lack of control).

## 2.3. SAV SP Typologies and Attributes Considered

Among SAV/public transport stated preference studies, studies have focused on different aspects and types of AVs, including general use of SAVs [36,40,46], demand-responsiveness [30], SAVs as first mile/last mile modes [49], and the presence of safety drivers [47]. These choice studies focus on using SAVs as a mode choice for trips.

Five of the above-mentioned studies employed logistic regression model techniques, including nested logit [36,46], mixed-logit [30,40,47,52], and nested mixed-logit [49]. Among SPs that focused on privately owned SPs, logit [32,33,38,41,44,46], mixed-logit [34,35,37,42], or logit kernel [43] were also commonly employed modeling techniques. Modes competing with AV in SP studies have included rail and bus transit [30,36,40,47,49,52,53], walking [36], non-automated private vehicles [36,40,49], non-automated taxi [36], and bicycles [36,40,49,53]. Among studies considered in this review, service characteristics examined included cost [30,36,40,49], in-vehicle time [30,36,40,49], waiting time [30,36,49], access or egress time [49], driverless or not [40,47], shared or not [40,49], and comfort [53].

## 2.4. SAV Studies and Findings

Krueger et al. [30] utilized an SP survey of respondents in Australia (*n* = 435) to analyze the characteristics of users who are likely to use SAVs, and determined willingness to pay measures for service attributes (travel time and costs). In this survey, respondents were given a choice of three alternatives: an AV service without sharing (i.e., robo-taxi), a SAV with sharing (i.e., automated public transport), and the current existing public transit network. Attributes included travel cost (in AUD), door-to-door travel time, and waiting time (i.e., time spent not moving). A mixed-logic model was employed. They

determined that the most important service attributes were travel time, waiting time, and fare, respectively. They determined that multi-modal travelers would be most likely to adopt an SAV, whereas those who used private cars would be less likely. Additionally, they found that younger respondents (24–29) were more likely to choose a demand-responsive SAV service.

The stated-preference-based study by Yap et al. [49] focused on AVs as a first-mile/last-mile transportation alternative in combination with longer-distance train journeys. Respondents from the Netherlands (*n* = 761) were given three alternatives to choose from: access/egress by AV and train, access/egress by public transport and train, and finally access/egress by bicycle and train with some variants. Journey attributes include in-vehicle time, waiting time, and travel cost. The authors employed a nested mixed-logit model. They concluded that AVs have good potential to be used for first mile/last mile services, but that respondents on average associated more disutility to the in-vehicle time in an AV, or in other words, slightly more people preferred to manually drive rather than use an SAV.

A similar result to the above was found in an Italian study undertaken in 2019. In this study, results indicated respondents would pay more (up to €2.31/trip) and travel longer (up to 9 min/trip) instead of using SAVs [52].

Sweet [40] conducted an SP survey of residents in the Greater Toronto and Hamilton Area (GTHA), Canada (*n* = 1684), and applied mixed logit models to estimate consumer interest in adopting on-demand ride railing, shared on-demand ride-hailing, AV on-demand ride-hailing, and shared on-demand AV shuttles. The sample was diverse and closely matched the actual demographics of the GTHA. Choice alternatives included driving, cycling, on-demand vehicle, public transit + on-demand vehicle, or public transit. Two models were created: a mixed logit 'base' model and a mixed logit model with mobility controls. Sweet concludes that privately operated automated vehicles (TNCs and individually owned) will have the best opportunity for success, and public transport automated vehicles will be less successful, and could be at a 'significant disadvantage' compared to the private sector.

### 2.5. Safety and SPM

By its nature, shared personal mobility (SPM) subjects the user to certain risks that could cause injury or even death. SPM, particularly electric scooters, have been associated with serious accidents. Farley et al. [18] write that, in a five year period between 2014–2019 in the USA, there were more than 70,000 accidents involving electric scooters, with 27% causing head injuries. Related research from Ma et al. [19] concluded that electric scooter injuries represent an emerging public health issue. Injuries can occur not only for users, but also for other pedestrians [20]. Risk factors for accidents involving bicycles, including electric bikes, are also well-known, and enhanced or dedicated bicycle infrastructure, including bike lanes, have been known to decrease the risk associated with riding a bicycle on a road shared with cars [25,26]. Studies from Israel have shown that electric bicycle safety does differ somewhat from that of the non-electric version, and while users of electric bikes have a lower injury rate compared to regular bikes, they have a higher risk for serious injuries, including head injuries [21].

### 2.6. Gaps in the Literature

While there are numerous studies focused on AVs using SP including autonomous shuttles/SAVs, as well as SP studies focused on shared mobility products themselves [54], there are no studies that directly compare two of the new mobility modes: shared personal mobility (SPM), and a shared autonomous vehicle. In university campuses or similar environments, SPM systems are already common [17,55], and SAV pilot projects have been implemented and are predicted to be used in campus-like environments in the future [7,29,56]. While some studies have undertaken intention-to-use and stated-preference-type analyses of shared electric bicycle systems [54,57], AV modes were not alternatives in these studies, and it is therefore not well understood what factors would affect the choice of these modes

when in competition with each other. In addition, there are few studies in the literature that examine the effects of comfort/convenience for a SAV, or conversely, safety and availability/convenience for shared bicycle system (SPM). As such, it is important to understand the influences of these two emerging and potentially complementary mobility services on mode choice, and further understand how passenger comfort, convenience, safety, travel times, and costs play a role in selecting each mode.

## 3. Study Area

The KNUT Chungju campus is located approximately 5 km west of the city center of Chungju City, North Chungcheong province, Republic of Korea, about 120 km southeast of the capital, Seoul. As with many universities in Korea, this campus is built on the side of a small mountain, and thus most roads have steep grades; refer to Figure 1. Walking between buildings can be onerous. Within the campus, there exists one public bus, route 999, which connects the main Chungju bus terminal to the top of the KNUT campus; 90% of bus users surveyed indicated that they used this route. This route takes around 15–20 min from end to end and runs approximately every 30 min on days when school is in session. Numerous other bus routes serve the Korea National Route 3 stop near the campus, but this can be a 15–18 min walk, depending on the start/end point on campus. The 999 bus is occasionally used for short-distance trips (14% indicated they would use it for short-distance trips). More popular methods of transport within the campus include walking (77% indicated they walked), SPM (25% indicated they regularly used these systems), and car (22.5% indicated they sometimes drove). When asked why they walked on campus, around 50% of respondents indicated it was because there was no other reasonable service, 8% because it was a substitute for another service, and 31% for their personal health, and 11% for other reasons.

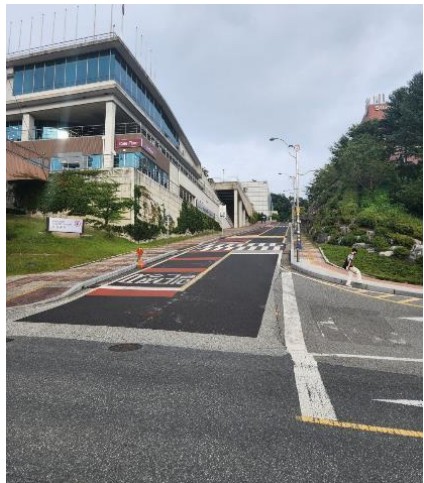 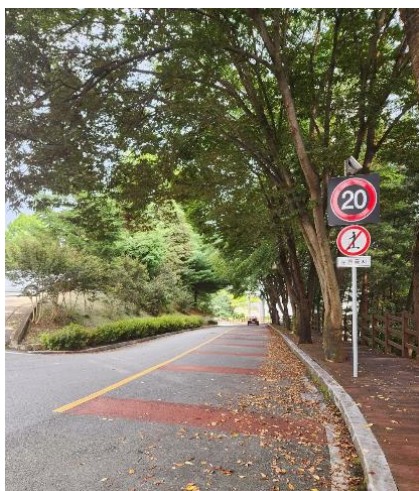

**Figure 1.** Examples of steep roads on campus.

The target population for this study are individuals who have the opportunity to use Ohmio, a Society of Automotive Engineers (SAE) Level 3 [58] autonomous shuttle which was demonstrated on campus, as well as the SPM that is available on campus. In other words, any individual registered with the university as being an undergraduate or graduate student, faculty member, or employee of the university's Chungju (main) campus. The target population was about 7360 individuals [59].

### 3.1. Ive and Ohmio SAVs

The KNUT Chungju campus has been host to two different autonomous shuttle demonstration projects. The most recently implemented demonstration is an autonomous shuttle demonstration built by New Zealand-based Ohmio and was designed to be an SAE Level 4 vehicle [60]. This demonstration officially started operating on the KNUT campus

in December 2021 and operated until February 2022. Previously, a Level 3 shuttle called Ive operated on the KNUT campus between 2017 and 2019. Ive is guided by a magnetic strip embedded in the roadway, and has some limited functionality in terms of sensing ability, while Ohmio is a more sophisticated vehicle that primarily uses light detecting and ranging (LiDAR) and global positioning systems (GPS) to drive autonomously. As is required for all AV vehicles in Korea, a safety driver was present in the shuttles when in operation. Further demonstrations or pilot projects are planned for the future. Note that both these pilot projects were technology demonstrations only, and did not reflect the same level of service as for the SAV service implied in the survey for this study (Figure 2).

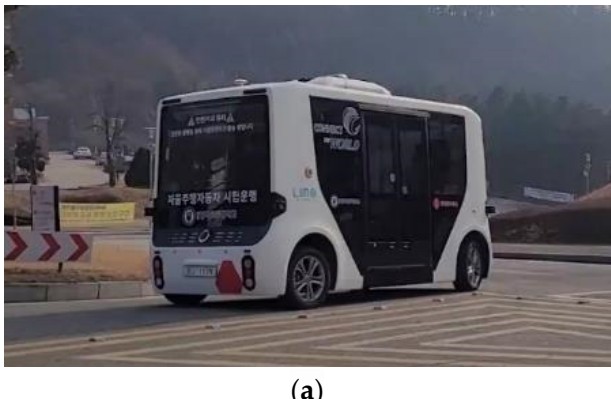
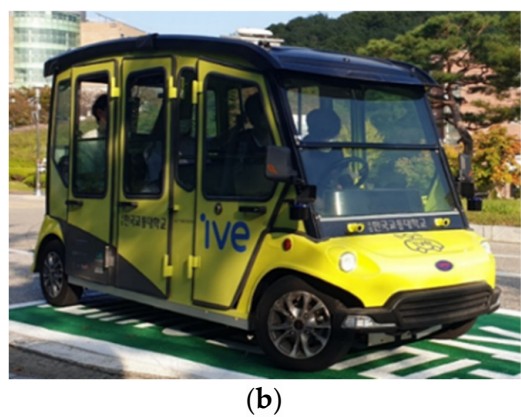

(**a**)　　　　　　　　　　　　　　　　　　　　　　　　　　(**b**)

**Figure 2.** Ive and Ohmio autonomous shuttles. (**a**) Ohmio (2021); (**b**) Ive (2017–2019, 2021).

*3.2. Shared Personal Mobility (SPM)*

There are currently two competing electric and shared modes of transport on the KNUT campus: a shared electric bicycle system (Elecle) and a shared electric scooter (kickboard) system (Deer). Both Elecle and Deer are privately owned SPM companies, and their systems are available in cities across Korea. Both Elecle and Deer have been in place and in use on the KNUT campus since September 2021 and were available for survey respondents to use during the survey period (December 2021–January 2022). To use Elecle or Deer, users must download an application on their SmartPhone and input credit card or banking information. Elecle bikes and Deer scooters placed in their 'drop zones' are shown below in Figure 3. Both have proven to be popular modes of transportation on the KNUT campus, especially for trips going up the campus' very steep sections. Elecle was initially operated free of charge, provided by KNUT, but a conversion to a paid service was implemented on 1 March 2022. Elecle now costs 400 Won ($0.29 USD) initially, plus 150 Won ($0.11 USD)/min. In a four-month period from March–June 2022, Elecle was used at the Chungju campus more than 68,000 times, with as many as 1605 individual trips per day. Average trip time was 4.48 min, which would imply an average cost of approximately 1073 Won ($0.78 USD) per trip (actual revenue and cost/trip were not provided by the operating company and are an estimate only). Deer operates with a similar cost structure as Elecle: 790 Won ($0.57 USD) initially, and then 150 Won ($0.11 USD)/min. Various discounts and promotions are also available. Usage and revenue data were not available to us for the electric scooter system, which is not officially endorsed by school administration and operates only semi-legally.

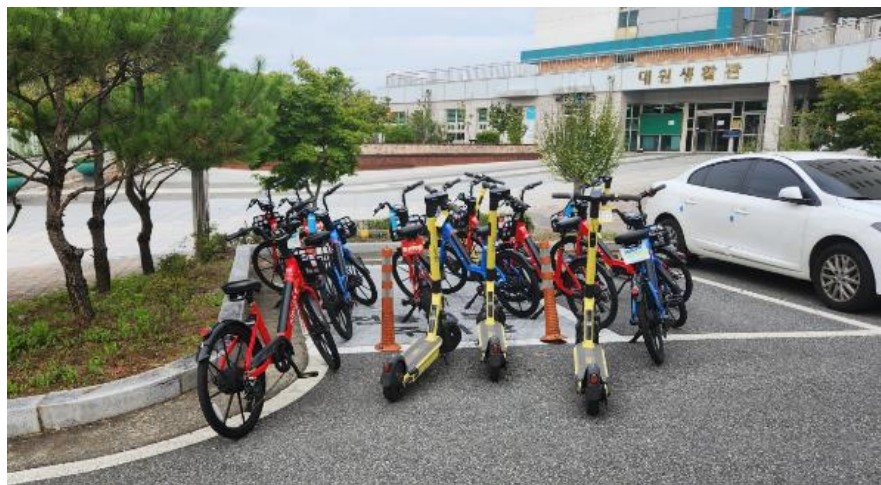

**Figure 3.** Shared personal mobility at the Chungju Campus (shared electric bicycles & shared electric scooters).

### 3.2.1. Incidents and Accidents

Throughout the operating history of Ive (2017–2019, 2021) and Ohmio (2021), no major accidents or incidents were reported. Between 2017–2021, Ive operated for more than 2072 h and Ohmio for 52 h. These vehicles operate at a low speed with a safety operator onboard ready to intervene in any emergency situation. However, since the shared electric bicycle and shared electric scooter (SPM) systems started operating on the Chungju campus in September 2021, there have been three reported non-fatal accidents, as follows:

- Major incident where an electric bicycle user collided with a motorcyclist
- Minor incident where an electric bicycle user fell over after going over a speed bump
- Major incident where an electric scooter user fell while going downhill, which resulted in hospitalization

In response to some of these incidents, especially the major incident involving the electric scooter, further safety measures were implemented. Speed warning signs (implemented on 15 February 2022) are in six places, and traffic safety signs (implemented on 28 February 2022) are in 25 places. Further, electric kickboards are banned from being used on some downhill sections. The speed limit was set to 20 km/hour for all road users (including SPM); refer to Figure 4 below.

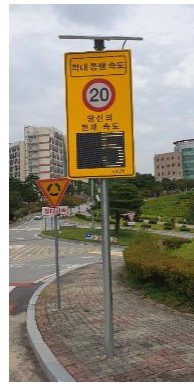
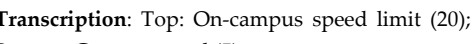
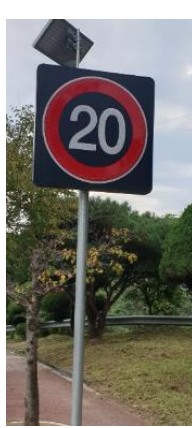
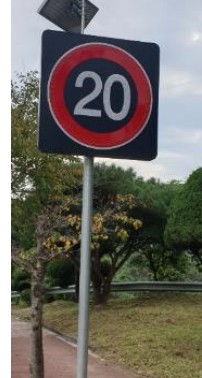
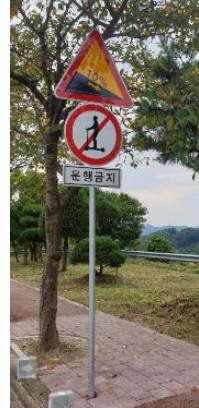

**Transcription**: Top: On-campus speed limit (20); Bottom: Current speed (5)

**Transcription:** Operation prohibited

**Figure 4.** Safety signage at the Chungju campus.

### 3.2.2. Actual Demand

Demand data for the Elecle shared electric bicycle system on the Chungju campus were made available to us for the period of 13 September 2021–13 September 2022. The data include information for every trip taken on Elecle, including time and origin-to-destination location. These data are shown in Figure 5, where trips are aggregated by week. During this time, demand was highly seasonable and variable for various reasons. During periods where class was not in session, demand was reduced to as low as just 39 weekly trips (during the Seollal holiday, which is a major 3-day holiday in Korea) to as high as around 6400 trips per week when classes were in session.

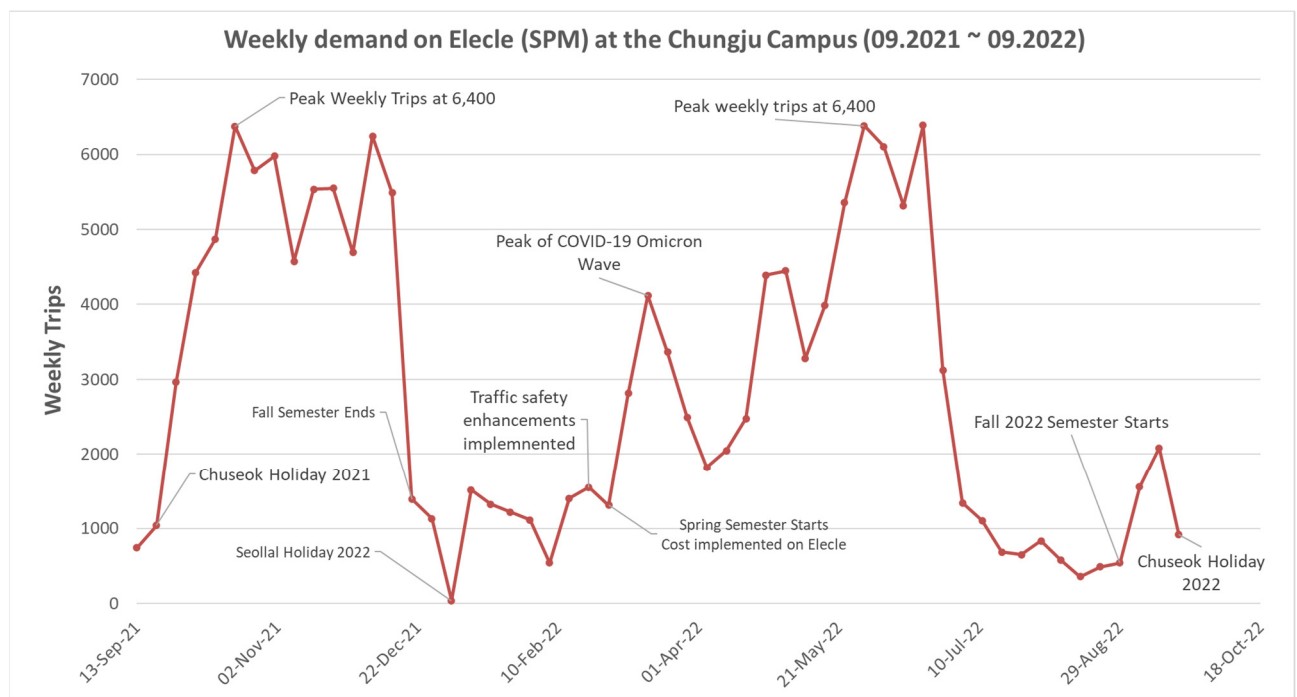

**Figure 5.** Weekly demand (trips) on Elecle (SPM) at the Chungju Campus (September 2021–September 2022).

Additionally, during this period, the cost structure of Elecle was changed from being entirely free to costing an average of around 1000 Won ($0.78 USD) per trip. This change occurred on 1 March 2022. In addition, some enhanced safety signage was installed in late February 2022. Overall, when comparing the core 14 weeks of the Fall semester (up to and including the exam period week (the first two weeks of both the Fall and Spring semester were omitted as data were only available starting on 13 September 2021) to the Spring semester, there was less frequent use of the system, with 13% fewer trips overall during this period. While this may be due in part to the cost increases of the system, given the highly variable nature of the data combined with other factors, we were not able to conclude whether this was the case, nor could we effectively model this demand or calculate elasticities.

## 4. Methodology

### 4.1. Survey Design

The survey was administered to potential respondents by email using the Google Forms platform. All faculty, staff, or students who were registered at the Chungju campus were invited to participate in the survey. The survey consisted of three parts: (a) general transportation habits and demographics; (b) the SAV acceptance survey used for this analysis; and (c) a stated-preference survey. After part (b), but before part (c), demographic questions were asked, including questions pertaining to age, gender, general residence location, relationship with the university (i.e., student, staff, or faculty) and income. Based

on an initial convenience sample pilot with 28 respondents, it took about 10–15 min to complete the entire survey. An incentive was given to respondents to the final survey in the form of a coupon good for one free coffee-based drink at a local coffee shop. The survey was undertaken in the Korean language; English translations are provided for this paper.

### 4.1.1. Pilot Survey

The purpose of the pilot survey was to confirm the validity of the selection of selected policy variables and the appropriateness of the range of changes in the policy variables chosen in advance and reflect them in this survey and analysis. The results from the pilot showed that both SAV and SPM variable coefficients were all significant with $p < 0.001$, and both the signatures of the coefficient estimates were consistent with our dictionary expectations, namely, time- and price- or cost-related variables were negative, and all variables related to convenience were positive. In the pilot survey (and main survey), an analysis was conducted only for those who responded that they would consider using SAV and SPM as possible options for a trip on-campus.

### 4.1.2. Sample Size and Descriptive Statistics

The survey accepted responses between 3 December 2021 and 13 January 2022. During this period, we received 248 responses to our survey; however, many were removed for the following reasons: 10 responses were removed initially as double (or triple) entries with the same email address; 39 respondents indicated that would not be willing to use an AV shuttle or shared electric bicycle, and were therefore excluded; and finally, 61 had biased/incomplete responses (i.e., they chose the same answer for every question). In total, 138 valid responses remained. The removal of the 39 respondents who indicated that they would not consider using the SAV or SPM as a viable transport option was to ensure that conclusions could be made from the results in a realistic SP analysis setting for individuals would actually use SPM and a SAV.

This represents a sample size of approximately 1.9% of the target population, or a margin of error of 7% for a 95% confidence interval. On average, respondents were 26.7 years old (only the year in which the respondent was born was available, so age was an estimate); 65% were male and 35% were female. In terms of their relationship to the university, 78% were students, 6% were faculty, and 16% were staff/other. The make-up of the sample is shown in Table 1, and is compared with data received from university administration for the 2021 target population.

**Table 1.** Sample/population demographic characteristics.

| | Sample | | KNUT Population | |
| --- | --- | --- | --- | --- |
| | Number | % of Total | Number | % of Total |
| Total | 138 | 1.9% (of population) | 7360 | - |
| Students | 108 | 78% | 6459 | 88% |
| Faculty | 8 | 8% | 546 * | 7% |
| Staff | 16 | 12% | 355 | 5% |
| Other | 9 | 4% | N.A. | - |
| Male | 90 | 65% | 5213 | 71% |
| Female | 48 | 35% | 2147 | 29% |
| Born > 2001 | 8 | 6% | 1196 | 16% |
| Born 1997–2001 | 69 | 50% | 4117 | 55% |
| Born 1992–199 | 31 | 22% | 851 | 11% |
| Born 1987–1991 | 7 | 5% | 133 | 2% |
| Born 1982–1986 | 6 | 4% | 162 | 2% |
| Born < 1982 | 17 | 12% | 1016 | 14% |

* Value includes 'faculty' and 'other'.

### 4.1.3. Respondents' Profiles

Respondents were asked a variety of questions related their travel habits and access to transport vehicles and their attitudes towards SAVs and SPM. Respondents were asked to indicate the frequency of their visits to the Chungju campus during the Fall 2021 semester, and to indicate their travel habits in January 2020, before the COVID-19 pandemic started. Figure 6 below illustrates the frequency of weekly visits to the Chungju campus by respondents in early 2020 (before any effect from the COVID-19 pandemic) and 2021 (during the COVID-19 panic and subsequent travel disruptions). Approximately 64% of the respondents visited the campus at least five days a week in 2021, while about 75% visited the Chungju campus more than five days a week in 2020, down about 11% from the previous year. However, results indicate that despite the pandemic, the respondents still most commonly commuted to campus 5 days per week, although this portion decreased by more than 10% during that time. Approximately 38% of respondents lived on or very near campus, while 62% commuted to campus.

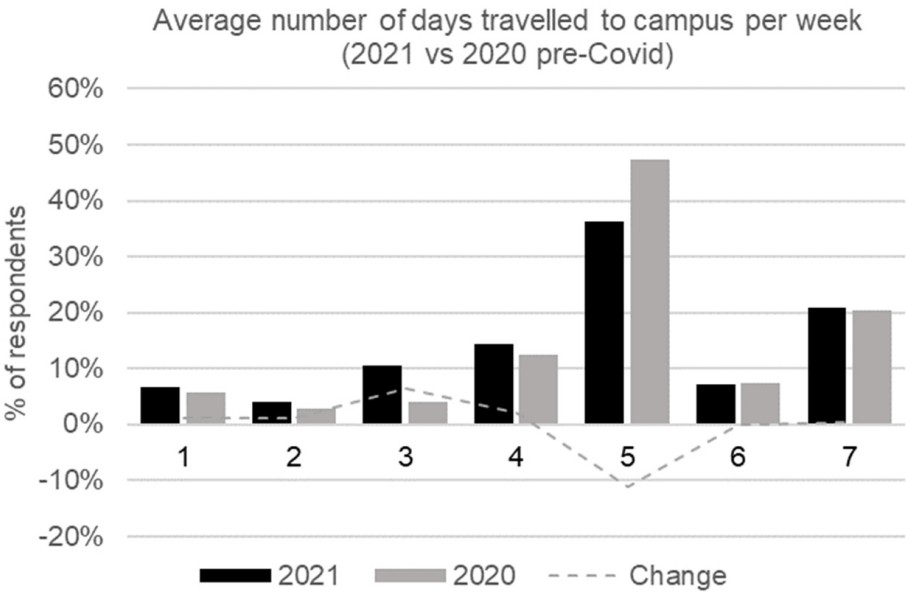

**Figure 6.** Travel to the Chungju campus (days/week).

Reflecting the many students who lived on/near campus, most respondents commuted on foot (45%) and 26% used a car and/or motorcycle, while public transportation was the preferred mode of 19%, and shared personal transport, such as electric bicycles (or electric kickboards), was preferred as a means of commuting by 7% of respondents. Additionally, nearly half (49%) indicated that they did not have access to personal transportation in their home. About 36% of the respondents had access to a car, 28% had access to a bicycle, and 7% had access to an electric scooter. Around 1% had access to a motorcycle.

Respondents also indicated their experience with and their willingness to use the transport modes that are the subject of this SP survey: shared autonomous vehicles and shared electric bicycles. The data are shown below in Figure 7. Approximately 44% of respondents had used a shared electric bicycle, while about 13% had used the autonomous shuttle (at the time of the survey, the shuttle had only been running for a few weeks, and many had not yet had the opportunity to use it). However, 88% of respondents indicated that they would be willing or may be willing to use a shared electric bicycle, and 98% indicated that they would be willing or may be willing to use the autonomous shuttle. Respondents who indicated they would not be willing to use either of these modes were excluded from the analysis.

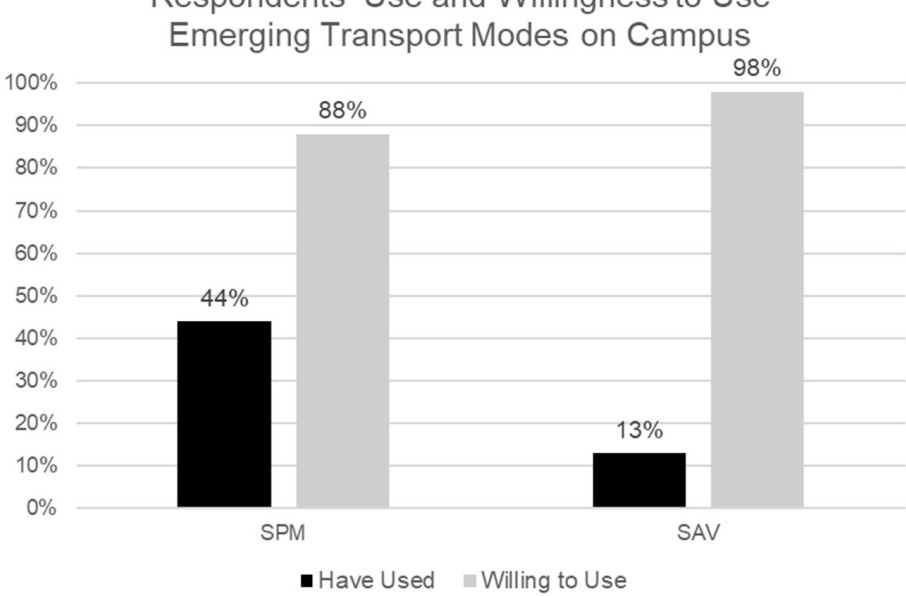

**Figure 7.** Experience and willingness to use an electric bicycle and autonomous shuttle.

Major differences between genders were not observed in the data, except for the availability of a car for driving. While only about 61% of men indicated that they did have a driver's license or availability of a car, approximately 81% of women indicated that they either did not have a driver's license or did not have access to a car in their household. However, overall commute modes were largely similar, with the most common method of commuting being on foot for both men (49%) and women (43%); see Figure 8 below. However, some notable differences were apparent with car and SPM. Respondents were also asked about the frequency of SPM usage per week. While major differences were not noted, with 48% of both women and men reporting not using SPM regularly, a larger portion of women (17%) used SPM regularly (more than 3 times per week) than men (13%). More women (11%) than men (6%) used SPM for daily commuting.

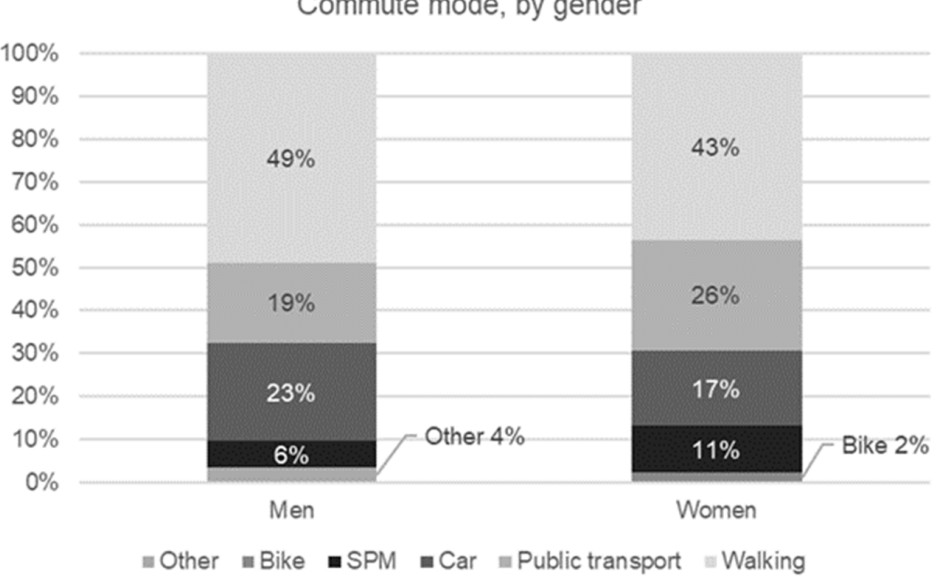

**Figure 8.** Commute mode, by gender.

Respondents, when asked if they trusted autonomous vehicles, expressed an average value of 3.35 on a 1–5 Likert scale. Further, respondents rated their knowledge of au-

tonomous vehicles at 3.37, and their knowledge of the introduction of autonomous vehicles on the campus as 3.90, also on a 1–5 Likert scale. Overall, the profile of the sample indicates a diverse set of individuals, despite nearly 80% being students, with a mix of those who lived on campus and off, with a significant portion of the sample willing to use the electric bicycle or autonomous shuttle. However, it should be noted that the sample and study area had some unique characteristics. The sample was primarily composed of students attending a suburban university campus that had numerous steep roads, and respondents may therefore be willing to use motorized transport options over short distances, which would be more walkable in a flatter environment. Additionally, a quite significant portion of respondents expressed a high trust and knowledge of autonomous vehicles, and correspondingly 98% indicated they would be willing to use one.

*4.2. Stated-Preference (SP) Approach*

Stated preference (SP) methodology is a discrete choice analysis for quantitatively measuring the effects of currently non-existent traffic services or policy measures [22,61,62]. SP methodology estimates the respondent's utility function through selection data related to various attributes of transportation on the premise of a hypothetical market situation, and quantitatively estimates the response of transportation users to policy variables from the estimated utility function. Specifically, the application is flexible and is extensively used in transport studies [22]. KNUT's planned introduction of SAV services can be most adequately analyzed by this methodology. The specific application procedure of the SP methodology is summarized step by step below.

1. Identifying a set of attributes, then
2. setting the unit of measurement for an attribute variable, then
3. setting the level of change in an attribute variable, then
4. experimental design, then
5. survey design, then
6. model estimation, and finally
7. the response.

In particular, the experimental design process of the fourth step above exists only in the SP methodology, and is reconstructed into a set of alternatives by combining the attribute variables and levels required in the SP methodology [61]. Quantitative analysis is possible at a relatively low cost through a survey of relatively few samples of users. Additionally, SP methodology can minimize bias, because it is difficult for respondents to take strategic action through scenario selection on a virtually set questionnaire [61]. SPs have been widely used in transport studies, including for SAVs [50].

The questions and alternatives that were asked in the SP survey are summarized in Table 2 below. A total of 18 hypothetical travel scenarios were presented to respondents (respondents were randomly divided into two groups, each of which were asked nine SP-related questions), each with two options: SPM or SAV.

**Table 2.** Stated-preference choice experiments: alternatives and attribute levels.

| Attribute | Text | SPM | SAV |
|---|---|---|---|
| Cost | Cost (per trip) | Free, 500 Won, 1000 Won | Free, 500 Won, 1000 Won |
| Waiting time | Waiting time | N/A | 2 min, 5 min, 10 min |
| Perceived safety | Safety | Always safe, sometimes safe, always dangerous | N/A |
| Perceived convenience | Congestion (crowdedness) (SAV), bicycles available (SPM) | Always available, sometimes not available, often not available | Seats always available, sometimes need to stand, always need to stand |

### 4.3. Discrete Choice Model

NLOGIT 6's Discrete Choice software [63] was used for analysis using a standard binary logit model. In addition to estimating the coefficient of the utility function, the elasticities of demand were also estimated using the same software to obtain the degree of response to each attribute of the utility function. NLOGIT uses the full information maximum likelihood estimation method (MLE) for logit model parameter estimation. These models relax the assumption of independently distributed errors and the independence of irrelevant alternatives (IIA) inherent in logit models. Model fit parameters were examined including log likelihood, $R^2$, and adjusted $R^2$. The following equations describe the multinomial logit model used in the analysis of the SAV and SP modes; as previously discussed, it includes time-related, cost-related, perceived convenience-related, and perceived safety-related variables.

$$U_{SAV} = \alpha_{SAV} + \beta_{SAV\_F} \times SAV\_fare + \beta_{WT} \times WTime + \beta_{CONG} \times Cong \text{ (SAV)}$$

$$U_{SPM} = \beta_{SPM\_F} \times SPM\_fare + \beta_{SF} \times Safety + \beta_{ES} \times Avail \text{ (SPM)}$$

where

$U_{SAV}$ = utility function of the SAV mode,
$U_{SPM}$ = utility function of the SPM mode,
$\alpha_{SAV}$ = constant term of the SAV (Indicating the effect that other variables used in the estimation cannot capture),
$\beta_{SAV\_F}$ = fare (charge) for SAV,
$\beta_{WT}$ = wait time for SAV,
$\beta_{CONG}$ = perceived convenience of SAV (in-vehicle congestion),
$\beta_{SPM\_F}$ = fare (charge) for SPM,
$\beta_{SF}$ = perceived safety of SPM, and
$\beta_{ES}$ = perceived convenience of SPM (bicycles available to use).

The price variable in the estimating equation is the SAV's fare, and the time variable included is only the waiting time. Fares are hypothetical fares, since no fares are imposed currently, but later impositions are planned. The responsiveness of users is our concern prior to actual charge in the near future. Driving (in-vehicle) time is not particularly important for this SAV, since it only operates on a limited area and is not likely to increase its driving speed due to technical and legal limitations. Further, waiting time is known to be more important to public transport users than in-vehicle time [64]. A variable for the fare of the SPM, which is currently provided free of charge, is also included. In the future, the Elecle bicycle share system may have a user charge.

Besides cost and time variables, perceived convenience and perceived safety variables were considered in this analysis. For the SAV, perceived convenience refers to the in-car congestion, that is it is a measure of how crowded an SAV vehicle's interior is. For the SPM, perceived convenience is represented by the availability of bikes for use, which is especially relevant given that the system is well-used, especially during peak times. In the survey, convenience (availability) is represented by the following choices: (1) always available, (2) sometimes not available, and (3) often not available. Perceived safety is applicable to the SPM (electric bike share) only; it is explained to the respondent that it represents the safety on the road, including whether there are fast-moving cars nearby or bicycle lanes provided.

## 5. Results

When this estimation model was applied, the following utility function estimation parameter results were obtained; refer to Table 3 below. All coefficients were statistically significant with $p < 0.001$. McFadden's pseudo-$R^2$ value [65] of 0.17 was in line with other recent SP studies involving SAVs [40].

**Table 3.** Estimation of coefficients of utility functions (main survey).

| Variable Name (Long) | Variable Name (Short) | Applies to | Coefficient | t-Value |
|---|---|---|---|---|
| Log likelihood | −955.1726 | | | |
| McFadden's $R^2$ | 0.17 | | | |
| SAV (alternative specific) constant | SAV_constant | SAV | 2.5753 *** | 7.99 |
| SAV fare | SAV_fare | SAV | −0.00173 *** | −11.46 |
| SAV waiting time | WTime | SAV | −0.0986 *** | −5.18 |
| Perceived convenience (congestion) | Cong | SAV | 0.376 *** | 4.98 |
| SPM (Elecle) fare | SPM_fare | SPM | −0.00124 *** | −8.18 |
| Safety | Safety | SPM | 0.364 *** | 4.85 |
| Perceived convenience (availability) | Avail | SPM | 0.632 *** | 8.14 |

*** $p < 0.001$.

All the coefficients were found to be of the expected value (positive versus negative). The positive and significant constant term for SAV implies people's intrinsic preference of SAV compared to SPM. Similar results to the pilot analysis were found in the main analysis, and the values of the cost and time coefficients were also negative, while the coefficients related to convenience and safety were also positive. The utility function of SAV included a constant term expressed by $\alpha_{SAV}$, which can be interpreted as a combination of the effects of all other variables that the time-related, cost-related, and convenience-related variables did not represent; this represents a fundamental (intrinsic) preference for the SAV.

### 5.1. Value-of-Time

From the above utility function estimates, it is possible to estimate the user time value of a transportation means having a time-related variable in the utility function. The value of time (VoT) of the SAV user may be expressed by the following equation.

$$\text{VoT}_{SAV} = \beta_{WT} / \beta_{SAV\_F} \times 60$$

In this case, the user's time value was estimated to be 3420 Won ($2.50 USD) per hour for the SAV. This was lower than other SP studies involving SAVs from Canada ($6.50–$7.36 USD) [40] but was similar to previous studies from Germany ($2.00–$3.00 USD). In this case, a low value of time may be reflective of the high student proportion of this sample, who generally had low incomes.

### 5.2. Elasticities of Demand

To estimate the degree of response to the effect of the attribute of the utility function in transportation method selection, the demand elasticity of each variable of the selection probability was estimated, as shown in Table 4 below.

**Table 4.** Estimation of demand elasticities.

| Variable Name (Long) | Applies to | Elasticity |
|---|---|---|
| Fare | SAV | −0.45 |
| Waiting time | SAV | −0.25 |
| Perceived convenience (congestion) | SAV | 0.42 |
| Fare | SPM | −0.42 |
| Perceived safety | SPM | 0.29 |
| Perceived convenience (availability) | SPM | 0.72 |

Price-related variables were estimated to be −0.45 and −0.42 for SAV and SPM, respectively; demand was found to be somewhat inelastic to fares in this case, although within the range of reported elasticities for bus-based non-SAV public transport, which is generally in the range of −0.28 to −0.55 [64,66]. The fare elasticity was estimated to be higher for SPM than for SAV, and implied that a fare system for the SAV may be feasible. For SPM, operators should understand that higher fares will indeed reduce potential demand for the service. Other studies have found elasticities for price in relation to demand for

shared bicycle systems in the range of up to −0.8 for low-income users [67]. In addition, in the case of the SAV, the elasticity estimate with respect to the waiting time was −0.25, which was more inelastic than expected, and may be due to the short nature of the on-campus trips. For non-automated public transport, elasticities in the wide range of 0.32 to 0.80 (for service) have been reported for buses [64].

A convenience-related variable was included for both SAV and SPM. In the case of SAV, convenience refers to the in-vehicle congestion, and for SPM (shared electric bicycles), the availability of bicycles. Perceived convenience-related elasticities for SAV and SPM were estimated to be 0.42 and 0.72, respectively. Results show that perceived convenience is more important in the case of SPM versus SAV, and this may reflect the characteristics of short-distance campus trips. That is to say that since the trips within the campus area were always relatively short, variables related to the vehicle journey (interior space congestion) were less important than the availability of bicycles for the SPM. In a setting where the trip journeys were much longer (for example from the university campus to the city center), respondents may have decided that in-car congestion was more important. With SPM, the availability of bicycles was not related to the trip length.

Finally, in the case of SPM, the elasticity of safety was also estimated to be 0.29. While this value is relatively low, it is more elastic than the value for waiting time for SAV. This value does confirm previous research, which has found road safety to be of importance for bicycle usage [26,54]. Estimations of time and cost, safety, and convenience-related variables in this study were generally consistent with previous studies, although slightly different in some ways, which we believe reflects the characteristics of these new transport modes as well as the short-distance nature of the on-campus trips.

## 6. Discussion and Conclusions

Implementing an autonomous vehicle service is one of the most important focus areas of KNUT to increase mobility on its Chungju campus. At KNUT, a pilot project of an advanced shared autonomous vehicle (SAV) was demonstrated in late 2021/early 2022, and in previous years, a more basic SAV (Ive) was also demonstrated, and provided service to university students and staff with a basic fixed route for approximately two years. During all the demonstration periods, which account for over 2100 h of service, no major or minor accidents were reported.

Additionally, the university supported an electric bicycle share system called Elecle, a type of shared personal mobility (SPM), and fully subsidized the cost of this service for several months before it was converted into a paid service. However, given the safety issues that have been found with SPM, and particularly electric scooters [18–20], the SAV service is preferred for the improvement of mobility on campus. When a SAV service is implemented, it will directly compete with SPM.

In this study, we estimated the factors that affected the demand for an SAV service competing with SPM, and identified key variables and their quantitative impacts on the demand by applying SP methodology. The survey results indicated that, if fully implemented, both SPM and SAV would be well-used by the populace at KNUT, with 88% indicating they would be willing to use SPM, and even more surprisingly, 98% of respondents indicating that they would be willing to use an SAV service if available. Of course, this does not directly translate into actual use system, since despite its being freely available for many months, only about 44% of respondents had actually used SPM. Around 13% of respondents indicated they had used an SAV during the free demonstration phases from 2017–2019 or in late 2021–early 2022.

While traditional SP variables related to fares and waiting time were included in this study, we also considered perceived convenience and perceived safety-related attributes, and we showed that these did affect people's modal choice behavior. Our results indicate that in addition to cost and time, SAV uptake will be as dependent on the level of congestion (that is available space for passengers) within the vehicle itself as on waiting time,

and relatively more important than cost, which is in line with previous work aimed at understanding convenience in non-automated public transport in Korea [68].

For SPM, our results show that the availability of bicycles is a very important aspect for its use, as is safety (that is, how safe the road is where the SPM will be used). The elasticities related to convenience (in car congestion) as well as cost and time for the SAV were estimated to be considerably lower than both the perceived safety and convenience variables for SPM. These results confirm that the SAV will be competitive with SPM if it is fully implemented, given the extremely high number of respondents who indicated they would be willing to use it (98%) and the calculated intrinsic value of the SAV mode when compared to SPM. This is desirable, considering that SAV has, historically, been safer than SPM. However, final uptake will depend on price, waiting time, and the level of congestion in the vehicle. These aspects can be controlled by the operator, including the in-car congestion—which in addition to service frequency is also a function of the vehicle design—and space afforded to passengers. Compared to traditional buses, SAVs without diver cabs should be able to offer more seating or standing room for passengers. Further, to increase the uptake of SPM, increasing the safety of the roads to increase the perception of safety as well as the availability of bicycles should be considered before pricing.

Finally, after examining actual system demand on the Elecle SPM system, meaningful conclusions or comparisons to the SP analysis were not possible to make. While demand in the Spring semester (when Elecle was not free to use) was indeed 13% lower than the in the Fall semester (when Elecle was free to use), many other factors likely affected the demand on Elecle, including weather, the implementation of new safety signage and policies related to SPM use, natural system growth and awareness, COVID-19 and its effects (in Korea, the first Omicron-related wave peaked near the end of March 2022), and the prevalence of university 'membership training' (in Korea, Membership Training (MT) is an annual event where new university students take part in organized outings for sometimes days at a time. At KNUT, these generally take place in April), which takes place in early April. Nonetheless, by June 2022 demand had again peaked at around 6400 weekly trips which is similar to the peak demand period of the Fall 2021 semester.

Our results show an intrinsic preference for SAV versus SPM. When considering factors that affect the demand, in addition to traditional cost and time variables, the effects of subjective variables, such as perceived convenience and perceived safety on demand, is relatively important and should also be considered, especially when studying these emerging transport modes.

*Limitations of This Study*

The population sampled for this study comprised mostly university students or university staff. The sample may therefore have a bias towards younger individuals and individuals who are more frequent users of public transportation. Further, as KNUT is primarily a technical university, its population skews towards males. The KNUT population is, however, representative of a population that is well informed and knowledgeable about SAVs. Comparisons between genders revealed no major differences, although women drove less than men did in this case. Finally, the sample was relatively small, with only 138 valid responses, due to many responses being removed for various reasons. The sample of 138 represented approximately 1.9% of the population from which it was taken.

**Author Contributions:** Conceptualization, S.L. and H.K.; Formal analysis, S.L. and D.F.; Funding acquisition, J.K. and H.K.; Project administration, J.K. and H.K.; Supervision, H.K.; Writing—original draft, S.L. and D.F.; Writing—review & editing, S.L., D.F., J.K. and H.K. All authors have read and agreed to the published version of the manuscript.

**Funding:** This work is supported by the Korea Agency for Infrastructure Technology Advancement (KAIA) grant funded by the Ministry of Land, Infrastructure and Transport (Grant 22AMDP-C161756-02).

**Informed Consent Statement:** All subjects gave their informed consent before participating in this study.

**Data Availability Statement:** The data presented in this study are available on request from the corresponding author.

**Conflicts of Interest:** The authors declare no conflict of interest.

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
