# Peer review of "Shared Autonomous Vehicles Competing with Shared Electric Bicycles: A Stated-Preference Analysis"

_sustainability, doi:10.3390/su142114319_

Round 1

Reviewer 1 Report

This paper is trying to understand the factors which affect the use of shared autonomous vehicles and shared electric bicycles as competing modes on a university campus in Korea based on a stated preference analysis. The topic is interesting and attractive. However, the paper still needs significantly major revision to improve the quality of the paper. If these problems cannot be fixed, it will be rejected.

1. The meaning of the statement in sentenceShared Autonomous Vehicles (SAV) and Shared Personal Mobility (SPM), in this case a shared electric bicycle system) are both promising new means of transportation that are receiving very high attention around the world [1,2]. is ambiguous. Can you give a brief introduction or explanation of SAV and SPM?

2.In the sentence “All studies considered types of AVs as the main alternative of interest with competing alternatives of rail and bus transit modes [27,33,37,44,46,49,50], only walking [33], 129 private vehicle (not-automated) [33,37,46], taxi (not automated) [33], or bicycle 130 [33,37,46,50].” its meaning of the statement is unclear.

3. “The author’s employed a nested mixed-logit model with and concluded that using automated vehicles as first-mile/last-mile transport, for first-class train journeys, is valued higher than using existing public transport or bicycles. However, the opposite conclusion was made for second-class train journeys. Overall, they conclude that AVs have good potential to be used for first mile/last mile services, but that respondents on average associated more disutility to the in-vehicle time in an AV, or in other words, the willingness-to-pay is higher for an AV compared to manual driving. This is a similar results to an Italian study undertaken in 2019 which indicated a respondents would pay up to more (up to €2.31/trip) and travel more (9 min/trip) instead of using SAVs [49].” I do not understand the meaning of the statement above.

4. There are errors in the data of students, faculty, staff and other in the sample in Table 1. Male should be 90 and female should be 48.

5. In the sentence “Model fit parameters were examined including log likelihood,  , and adjusted  .” an explanation of  should be added.

6. How to judge which of SAV and SPM is better based on the data in Table 4, please give the judgment basis.

7.In the sentence “In addition, in the case of the SAV, the elasticity estimate with respect to the waiting time was estimated to be -0.4, which more inelastic than expected, and may be due to the short nature of the on-campus trips.” the data is inconsistent with the data of Table 4.

8. Why is the expression about  in equation 3 represented by  in SPM?

9.The abbreviation should be given the full name when it appears at first time, for example, TNC, SAE.

10. The writing and language should be improved, for example,

“The sample was diverse and closely matched the actual demographics of the GTHA In this survey, choice alternatives included driving, cycling, on-demand vehicle, public transit + on-demand vehicle, or public transit.” Why “In” with capitalizing the first word?

 Results show that perceived convenience is a relatively more important variable to consider when comparing SAV to SPM, and this may reflect the characteristics of short-distance on-campus trips. For short-distance trips” Should “For short-distance trips” be deleted?

Reviewer 2 Report

Very intersting article, I the research covers only a small sample, but is an interesting introduction to this type of analysis. I am possitively jealous that he has the opportunity to conduct this kind of research at his university.

Reviewer 3 Report

Dear authors,

Your work is a case study with original elements that can be used by the global research community, if they are elaborated a little more in your paper.

Please take into consideration the following comments:

A. A minor syntax and grammatical check should be conducted throughout the text. E.g. on lines 27-28 you start a phrase with comma and end it with a parenthesis. On line 218 you mention "15-18 walk" possibly missing the word "minute" in between.

B. In the introduction, please show more clearly the scope of the research and if the purpose is to introduce new variables in the question about the applicability and suitability of SAVs and SPM through your methodology (as stated in the introduction) or/and comparative analysis between SAVs and SPM (as stated in the lit. review section). Also, please justify why the comparison between the two systems is useful to the research.

C. In the introduction, the paragraph on lines 43-53 should be embedded in the section for the description of the study area and be erased from the introduction, replaced by a very brief reference to the study area.

D. In literature review (line 79), you should indicate that the review also includes shared personal mobility (non-automated).

E. In the pilot survey, since the share of 39 respondents out of 248 is significant, further examination on the characteristics of respondents and the reasons why they would not consider using the new services may be of importance to the study and should be analysed more.

F. In the respondents' profile, it may be interesting to examine the current mobility profiles also in correlation to age and gender, as it would offer an insight on the characteristics of the target groups for potential modal shift to the new services. 

G. In the SP approach, concerning the attributes description, please elaborate on how the cost attributes levels were assessed and explained to the respondents. It is observed that the costs are equal per/trip for different services in terms of operating costs, business models and trip/service characteristics. 

H. On lines 444-446, despite arguing that in-vehicle time (only in the SAV, not on the electric micromobility mode) is not crucial to the choice, the total travel time (waiting and in-vehicle time combined) with the new service can (should) be correlated with the current total travel time to examine willingness if the trip duration were shorter, the same or even longer to the current trip duration and by what portion. Total travel time vs. cost is a typical comparison for mode choice decisions.

I. In equation 3, possibly "bSF" should be "bSAV_F", if I understand correctly.

J.  Actual demand is not generated by the survey and is not correlated directly to the survey results. Thus, it should maybe be moved further up in the description of the study area.

Reviewer 4 Report

This paper is entitled “Safety and Convenience-Related Variables for Shared Autonomous Vehicles and Shared Electric Bicycles: A Stated Preference Analysis”. In this study it undertook a stated preference analysis (n=138) to understand the factors which affect the use of shared autonomous vehicles and shared electric bicycles as competing modes on a university campus in Korea. In this case, the idea and results of the paper are interesting but the following comments can be utilized to improve this paper in future.

Abstract,L11: It is better (n=138) remove.

Abstract: A brief information related to the prior problem must be described.

Line43 to 53: this information must be moves to the case study. In the part of the introduction, authors must provide introduction of the study (i.e. existing research).

Line 54: “In preparation for this, in this study, the factors affecting the demand for SAVs and”: If authors provide information related to the current research, why they mentioned citation “[17–19]” at the end of the paragraph.

Line 91 and 92: both of them are using citation [6]. It is better to use only one time.

Line148-162: Authors used citation [46] at the first part of the paragraph and citation [49] at the end of that. It is not correct and paragraph must be split into two parts.

Line 179: “They and others [52]” What is the meaning of the They and others. The name of the authors based on the format must be provided.

Figure 2: It is better to move to the prior page in the blank part.

The caption of the Figures within all parts of the manuscript must be check. The position of the captions is at the bottom of the figure.

Line 306: “Based on an initial 306 pilot with 28 respondents”: Why 28 respondent. In must be describe precisely based on the existing probability or statistic formulas.

Line 312: The purpose of the pilot survey was to confirm the validity of the selection of selected policy variables and the appropriateness of the range of changes in the policy variables chosen in advance and reflect them in this survey and analysis.: how it confirmed?!?!

Line 314: “The results from 314 the pilot showed that both SAV and SPM variables were significant at the 1% level in co-315 efficient estimation”: How it (1%) was calculated?!?!

Line312: “The survey accepted responses for a period of approximately one month in Decem-322 ber 2021 and January 2022.” : why two monthes and why “December 2021 and January 2022”?! It must be describe in details

Line 323: How do the authors understand this number is enough?! Where is the statistical formula for data collection?!

“138 valid responses remained”: if the final data coe from this number, how do the authors understand this is enough for their research.

Table 1 must be rearranged in one page. If it is impossible, it must be separate with two captions.

Line 426: Equation do not need any description. If authors need to describe, it must be performed in the text of the manuscript. This comment must be consider for other equations.

Final decision: The idea and objective of this paper are interesting. The structure of the paper is suitable and it can be published after major revision. 

Reviewer 5 Report

This study conducted a stated preference analysis to understand the factors which influence the use of shared autonomous vehicles and shared electric bicycles as competing modes on a university campus in Korea. Overall, the manuscript is in good shape and requires minor revisions.

1. The sample size of the data is small, and the rationality of the sample size should be explained.

2. The discussion on Table 3 is not sufficient and needs to be supplemented.

3. The limitation of this study should be discussion.

4.In Table 1, the sum of the numbers in the second row and the first column is not 138, please check this table.

5. Minor language typos should be corrected; lack of commas; long sentences should be rewritten (it is hard to follow).

Round 2

Author Response

Response to Reviewer 1 (Round 2) Comments

Thank you for your efforts to improve our paper.

Point 1: In the sentence “tive variables (convenience and safety) on the use of emerging transport modes is as important as traditional cost and time variables.” delete this blank line

Response 1: Corrected.

Point 2: There are two 2.2(“2.2 Stated preference studies involving AVs”/ “2.2 SAV SP typologies and attributes considered”). Please reorder the contents of the second part.

Response 2: Corrected.

Point 3: “3.3.2 Actual Demand” should be “3.2.2 Actual Demand”

Response 3: Corrected.

Point 4: In the sentence “In the future, when an SPM service is implemented, SPM and SAV will be directly competing services.. ” , be careful with the use of punctuation..

Response 4: Thank you. Corrected sentence as follows to be more clear and conscise:

When an SAV service is implemented, it will directly compete with SPM.

Reviewer 4 Report

no comments

Author Response

Thank you for your efforts to improve our paper.